# Virus Yellows and Syndrome “Basses Richesses” in Western Switzerland: A Dramatic 2020 Season Calls for Urgent Control Measures

**DOI:** 10.3390/pathogens11080885

**Published:** 2022-08-06

**Authors:** Mathieu Mahillon, Raphaël Groux, Floriane Bussereau, Justine Brodard, Christophe Debonneville, Sonia Demal, Isabelle Kellenberger, Madlaina Peter, Thomas Steinger, Olivier Schumpp

**Affiliations:** 1Research Group in Virology, Bacteriology and Phytoplasmology, Agroscope, CH-1260 Nyon, Switzerland; 2Research Group in Entomology, Agroscope, CH-1260 Nyon, Switzerland; 3Centre Betteravier Suisse/Schweizerische Fachstelle für Zuckerrübenbau, CH-3270 Aarberg, Switzerland

**Keywords:** sugar beet, *Arsenophonus*, aphid, *Myzus persicae*, *Aphis fabae*, polerovirus, closterovirus, neonicotinoïd, *Pentastiridius leporinus*

## Abstract

Massive outbreaks of virus yellows (VY) and syndrome “basses richesses” (SBR) are thought to be responsible for the major loss of sugar beet yields in 2020 in western cantons of Switzerland. Typical yellowing symptoms were visible during field inspections, and control measures were reportedly ineffective or even absent. Both diseases induce yellowing but have distinct etiologies; while VY is caused by aphid-transmitted RNA viruses, SBR is caused by the cixiid-transmitted γ-proteobacterium *Candidatus* Arsenophonus phytopathogenicus. To clarify the situation, samples from diseased plants across the country were screened for the causal agents of VY and SBR at the end of the season. *Beet yellows virus* (BYV) and *Beet chlorosis virus* (BChV) showed high incidence nationwide, and were frequently found together in SBR-infected fields in the West. *Beet mild yellowing virus* (BMYV) was detected in two sites in the West, while there was no detection of *Beet western yellows virus* or *Beet mosaic virus*. The nucleotide diversity of the detected viruses was then investigated using classic and high-throughput sequencing. For both diseases, outbreaks were analyzed in light of monitoring of the respective vectors, and symptoms were reproduced in greenhouse conditions by means of insect-mediated inoculations. Novel quantification tools were designed for BYV, BChV and Ca. A. phytopathogenicus, leading to the identification of specific tissues tropism for these pathogens.

## 1. Introduction

In Switzerland, sugar beet (*Beta vulgaris* subsp. *vulgaris*) is a major crop cultivated on 16,000–18,000 ha mainly distributed along the Central Plateau. Two factories located in Aarberg and Frauenfeld collect and process the harvests from western and eastern cantons, respectively. Since 2018, sugar yields have been much inferior in Aarberg, and this situation was particularly marked in 2020 (Figure 1). Two diseases, namely virus yellows (VY) and syndrome “basses richesses” (SBR, “low sugar content”), are thought to be responsible for this reduction.

VY is caused by several single-stranded, positive-sense RNA viruses: the closterovirus *Beet yellows virus* (BYV), the three poleroviruses *Beet mild yellowing virus* (BMYV), *Beet chlorosis virus* (BChV) and *Beet western yellows virus* (BWYV), and the potyvirus *Beet mosaic virus* (BtMV) [1]. The disease was first described in the 1930s in Europe, and was subsequently reported on other continents during the last century [2]. VY viruses are transmitted by different aphid species (superfamily *Aphidoidea*, order *Hemiptera*), among which the most efficient vector seems to be the green peach aphid (*Myzus persicae*) [1]. Other vectors have been evidenced, such as the black bean aphid (*Aphis fabae*), the bird cherry-oat aphid (*Rhopalosiphum padi*) and the potato aphid (*Macrosiphum euphorbiae*) [3,4]. Transmission modes vary among VY viruses; it is semi-persistent for BYV, persistent (circulative but not propagative) for the poleroviruses and non-persistent for BtMV [4,5]. VY has been efficiently managed since the 1990s in Europe by seed coatings containing neonicotinoids which allow for vector control [6]. Together with increasing insecticide resistance observed in aphid populations [7], the ban on neonicotinoids in the Swiss agricultural sector in January 2019 has led to a fear of re-emergence. During the 2020 season, cases of leaf yellowing were reported as early as the end of May in western cantons, suggesting that VY outbreaks occurred and calling for further investigation.

In parallel, the yellowing of sugar beet can also be associated with SBR, a disease caused by the γ-proteobacterium *Candidatus* Arsenophonus phytopathogenicus [8]. Besides reduction in sugar content and leaf yellowing, the distinctive symptoms associated with SBR are the narrowing of apical leaves and browning of root vasculature [8,9]. Ca. A. phytopathogenicus is an intracellular parasite of phloem sieve elements and is transmitted by the planthopper *Pentastiridius leporinus* (family *Cixiidae*, order *Hemiptera*) [10]. The bacterial pathogen is circulative and propagative inside its vector and it is also vertically transmitted to a third of the offspring on average [10]. First described in Burgundy and French Jura (East of France) in the 1990s, SBR has been reported afterwards in Germany [11], and since 2017 in Switzerland where it was initially detected in the western canton of Vaud and then in surrounding regions [12]. The spread of SBR is particularly alarming as there is no resistance in beet cultivars and no eco-friendly eradication method is available for *P. leporinus*.

In this context, a national survey was achieved in 2020 to screen plants exhibiting yellowing for the causal agents of VY and SBR. In parallel, monitoring data for the respective vectors were collected. Given their high prevalence, the diversity of BYV and BChV was assessed based on PCR amplicon sequencing. Furthermore, the first Swiss genomes for several isolates of BYV, BChV and BMYV were assembled. SBR and VY symptoms were efficiently reproduced in greenhouse conditions, which led to the identification of specific tissue tropism for the respective causal agents. This study concludes with a discussion on future control strategies for both diseases.

## 2. Results

### 2.1. Survey of VY and SBR in 2020 across Switzerland

In the course of the 2020 season, the yellowing of leaves was first reported at the end of May in the West of Switzerland. First noticed on small numbers of plants, these symptoms later spread as patches in infected fields (Figure 2A), which is a typical feature of VY [13]. Numerous aphids were visible on plants during routine inspections (Figure 2B). Spraying pirimicarb insecticides were applied but relatively late (i.e., June). Eventually, many fields in the West were completely yellowed at the end of the season. In addition, specimens of *P. leporinus* were observed on sugar beets during the summer in the West (Figure 2D) and distinctive SBR symptoms started to appear in August in most fields from the same regions (Figure 2C). 

Conversely, yellowing symptoms were noticed much later in the East, and were less widespread at the end of the season. No SBR-associated symptom was reported in these regions and the cixiid vector was not reported from field inspection.

To clarify the situation, leaf and root samples were collected from representative symptomatic plants in the main cultivation areas and subsequently tested for the presence of VY viruses and Ca. A. phytopathogenicus (Appendix A).

Out of 23 sites investigated for both diseases, only two in the East, namely Ellikon and Hagenbuch, were found negative for all five tested viruses and Ca. A. phytopathogenicus (Figure 3A,B), suggesting that another cause was responsible for the observed yellowing. BYV and BChV were frequently found across the country (Figure 3A). In the West, these viruses were detected in 80.0% and 83.7% of the samples, respectively, and the presence of both viruses was evidenced for 14 out of 20 sites. In the East, each virus was detected at a similar rate of 38.9%, but no site was found to be infected with more than one virus. BMYV was detected in three samples from two western sites, namely Gals and Moosseedorf, in mixed infection with BYV and BChV. All samples tested negative for BWYV and BtMV.

The PCR screening targeting Ca. A. phytopathogenicus revealed that 47.7% of the samples were positive in the West, corresponding to 11 out of 13 locations, always in association with at least one VY virus (Figure 3B and Appendix A). On the other hand, no sample from the ten investigated sites in the East tested positive for the bacterium. Since yellowing symptoms on sugar beet can also be associated with the Stolbur phytoplasma (*Ca.* Phytoplasma solani) [9], root samples were also screened for this pathogen by nested PCR using Stamp primers (Table 1). Only two samples from Changins were positive.

### 2.2. VY Vectors Monitoring

VY severity depends primarily on the incidence of overwintering aphids that can colonize reservoir plants and subsequently inoculate beets at an early stage [21]. Studies in the UK have revealed that the number of frost days in winter and the mean temperature in early spring account for the greatest percentage of the variance in VY outbreaks [22,23]. In Switzerland, the 2019/2020 winter was the warmest since the beginning of data recording in 1864 [24], and it was followed by a hot and dry spring [25]. In line with these meteorological data, the first alate specimens of two VY vectors were caught very early in 2020 at the Agroscope suction trap in Changins. Indeed, specimens of *M. persicae* and *A. fabae* were detected on the 9th and 16th of April, respectively (Figure 4), while both species have been detected around the 28th of the same month for the 2000–2019 period. In addition to this exceptional earliness, record-breaking abundances were observed. On the first of May, in the early phase of the growing season, cumulative aphid numbers were 36 times superior to the 2000–2019 average for *M. persicae* and 20 times superior for *A. fabae*.

### 2.3. Viral Sequence Analysis

Given the prevalence of BYV and BChV, their respective genetic diversity was explored in randomly selected samples throughout the country. To this end, short fragment located in the 5′ end of the genome from either virus was sequenced using the primers used for the survey.

In the case of BYV, three main haplotypes were identified (Figure 5A). The diversity of the sequenced fragment was rather low and there was no clear association between geographical distance and haplotype. Interestingly, one haplotype was identical to sequences obtained from Croatia in 2014 [26]. Globally, the Swiss haplotypes were more similar to sequences recently retrieved in the UK (BYV-PV-1237, unpublished) and Germany (BYV-PV-1260, unpublished) than sequences obtained decades ago from the USA (BYV-Ca and BYV-4 [27,28]) and Ukraine (BYV-U [29]).

As far as BChV is concerned, the diversity was lower than BYV as one major haplotype was identified (Figure 5B). This haplotype was highly similar to sequences obtained from the UK (BChV-2a [30]), France (BChV-PV-1211, unpublished) and Poland [31], and more divergent than sequences obtained from South Africa (BChV-MPTGP ZA, unpublished) and the USA (BChV-CR [30]). Interestingly, minor haplotypes found in the canton of Jura were quite distinct from the major haplotype and were more similar to sequences obtained from France. 

A high-throughput sequencing (HTS) analysis of virion-associated nucleic acids (VANA) was performed in order to obtain the genomes of BYV and BChV from eight leaf samples (Appendix A). While full-length BYV genomes were assembled with high coverages, this method did not succeed for BChV and Sanger sequencing was therefore necessary to fill the gaps. Eventually, three near-complete BChV genomes were compiled. In parallel, the three leaf samples that tested positive for BMYV were used to assemble near-complete genomes using classic sequencing. Accession numbers for these new genomes can be found in Appendix A.

The five Swiss BYV genomes were 15,470 nt in length and harbored termini similar to previously published genomes, advocating for a near complete sequencing. These genomes showed limited variability and grouped in two closely related clades (Figure 5C). Notably, the German BYV-PV-1260 grouped together with the genomes obtained from sites located near the German border, namely Ramsen and Dachsen, supporting a circulation between both countries. Sliding window analysis of nucleotide variation for all available BYV genomes revealed that the 3′ half accumulates more mutations than the 5′ half (Appendix A).

The three near-complete Swiss BChV genomes were also highly conserved and grouped together in a single clade related to genomes from the UK (BChV-2a) and France (BChV-PV-1211) (Figure 5D). Nucleotide variability across BChV genomes also revealed more mutations in the 3′ half, yet the pattern was less stringent than observed for BYV (Appendix A).

The three near-complete BMYV genomes were almost identical, sharing more than 99.9% nt identities (Figure 5E). As these genomes were obtained from samples in relatively close proximity, a common source can be hypothesized. These genomes accordingly grouped phylogenetically together (Figure 5E) and were found to be closer to French genomes (BMYV-2ITB [32] and BMYV-EK [33]) than to genomes from Germany (BMYVV-IPP [34] and BMYV-PV-1210, unpublished) and the UK (BMYV-Broom’s Barn [35]). The variation profile for BMYV genomes did not exhibit a clear pattern (Appendix A).

### 2.4. BYV and BChV Quantification Tools

A new RT-qPCR assay was developed to evaluate the titers of BYV and BChV and to allow for their simultaneous detection. Primers and probes (Table 2) designed within the sequences were used for haplotype analysis in order to take into account potential variation between viral genomes. In addition, both viruses showed low genomic variability in these regions as shown in the previous section. Based on the available sequences, 95% of isolates had no mismatch in any primer or probe target sequence and almost all divergent isolates differed by only one mismatch. When used in triplex, i.e., for the simultaneous detection of both viruses and the internal control, the sensitivity of the RT-qPCR detection was found to be 10^5^ times superior than the conventional RT-PCR for BYV, whereas the sensitivity was similar for BChV (Appendix A). The use of duplex detection improved the sensitivity of the test by a factor of 10 (Appendix A). 

These quantification tools were then used to assess the relative viral titers in different sugar beet tissues infected with either BYV or BChV using duplex reactions. To this end, plants of susceptible sugar beet were inoculated using viruliferous specimens of *M. persicae* in greenhouse conditions.

First symptoms were observed at 15 and 30 dpi for BYV and BChV, respectively, consistent with previous works [37]. Interveinal yellowing at early stages followed by chlorosis and small necrotic patches were visible on BYV-inoculated plants (Figure 6A-center,B-left). The symptoms developed on BChV-inoculated plants consisted of chlorotic patches at the tip of older leaves (Figure 6A,B-right). At 60 dpi, all plants exhibited leaf symptoms but no root symptom was noted.

Both viruses were detected by RT-qPCR in all analyzed tissues (Figure 6C), although titers ranged over two orders of magnitude. The highest viral titers were found in leaf blades. In the case of BYV, titers were higher in blades of emerging leaves than in blades of oldest leaves, while this pattern was opposite for BChV. Intriguingly, the latter was found to accumulate in rootlets at levels similar to what was found for leaf blades. No signal was detected in a similar analysis of healthy control plants (data not shown).

### 2.5. SBR Vector Monitoring

The cixiid *P. leporinus* has been identified as the main vector of SBR in France, but the other species *Cixius wagneri* was shown to be vector as well [9]. In order to clarify the situation in Switzerland, eight SBR-infected fields were visited in the 2020 and insects were collected via a suction device. Adults of *P. leporinus* were present at all sites, whereas no *C. wagneri* was collected. Insects were then tested for the presence of Ca. A. phytopathogenicus by PCR (Table 3). For this screening, the primer pair Alb1/Oliv1 was chosen (Table 1) as it produces a pattern of four amplicons for Ca. A. phytopathogenicus DNA, but not for closely related endosymbionts [18]. This analysis revealed a global infection rate of 64.4%.

### 2.6. Greenhouse Inoculation of SBR

Due to the lack of previous biological characterization of SBR in Switzerland, we aimed to reproduce the symptoms in greenhouse conditions. Twenty adults of *P. leporinus* were thus collected from a heavily infested field and were placed in insect-proof cages with ten sugar beets seedlings. Dead insects were collected after two weeks and were analyzed for the presence of Ca. A. phytopathogenicus by PCR, revealing an infection rate of 80% (data not shown). The first symptoms were visible at 52 dpi on one plant, consisting of deformed emerging leaves. At 90 dpi, taproot samples from four inoculated plants tested positive for the bacterium by PCR (Appendix A). These plants exhibited yellowing on old leaves, lanceolate and chlorotic emerging leaves and deformed taproots harboring brownish vasculature (Figure 7A,B). Notably, the four infected plants displayed significantly reduced taproot biomass in comparison to the healthy controls (Figure 7C). The remaining six plants were asymptomatic and similar to healthy controls. 

A novel qPCR tool was designed to estimate the bacterium titer relative to plant DNA in different tissues (Appendix A). This tool revealed high titers in petioles of old leaves, taproots and rootlets, whereas there was no detection in the blades of old leaves nor in the petioles or blades of emerging leaves (Figure 7D). No sample tested positive for the bacterium in a similar analysis on healthy non-inoculated plants (data not shown).

In parallel, ten adults of *P. leporinus* were used to inoculate four Madagascar periwinkles (*Catharantus roseus*) in similar conditions. Vein yellowing first appeared on one leaf from one plant at 40 dpi (Appendix A) and then spread onto other leaves (Appendix A). The bacterial DNA was detected in the stem (Appendix A, column “donor”). Four young periwinkles were subsequently grafted using stem parts of the infected plant. After three weeks, two grafted plants exhibited symptoms consisting of leaf yellowing and the bacterial infection was confirmed by PCR (Appendix A), confirming efficient transmission.

## 3. Discussion

This study showed that the 2020 season was marked with the re-emergence of VY in Switzerland. The preceding winter and early spring were characterized by meteorological conditions highly favorable for aphid development. These conditions resulted accordingly to exceptional earliness and abundances for the vector species *M. persicae* and *A. fabae* (Figure 4). However, the West and the East of the country experienced this re-emergence in very contrasting ways. In the West, field symptoms were reported earlier and were ultimately more widespread than in the East. At the end of the season, the majority of western fields were infected with multiple viruses, whereas these associations were not found in the East (Figure 3A). The reasons for such a geographical discrepancy are unclear and require additional examination. The presence of BYV, BChV and BMYV and the absence of BWYM in Switzerland reflects the situation encountered more globally in Europe, and in particular in neighboring countries [38]. Since these viruses have multiple alternative hosts [30,39] and can travel over long distances via aphids, VY will likely continue to be a serious threat on the whole continent. Even though BtMV was not detected in the survey, monitoring this virus remains relevant as it was recently detected in Germany and France [38]. It is worth mentioning that divergent variants might have escaped the RT-PCR screening, hence the non-detection of BWYM and BtMV should be taken with caution.

Haplotype data for BYV and BChV, the viruses representing the majority of VY cases in Switzerland, highlighted low variability and close proximity to sequences obtained from Europe in recent years (Figure 5A,B). These results were further supported by phylogenies of complete genomes (Figure 5C,D). Likewise, analyses of the three BMYV genomes indicated high conservation (Figure 5E). It is expected that additional sequences from other countries will help to better characterize the population dynamics of these VY viruses.

Symptoms of BYV and BChV were efficiently reproduced in greenhouse conditions following aphid-mediated inoculations (Figure 6A,B). RT-qPCR analyses of infected plants showed that leaf blades contained the highest viral titers (Figure 6C), proving that the sampling procedure was appropriate during the field survey. Although both viruses are restricted to the phloem [40,41], the tissue tropism was found to be markedly different; higher viral titers were detected in emerging leaves for BYV, while the highest loads were found in the oldest leaves for BChV (Figure 6C). Relatively high accumulation of the latter in rootlets was also evidenced and might be related to a particular susceptibility of this tissue as already reported for other viruses [42,43].

As expected due to typical symptoms visible in fields (Figure 2C), the national survey confirmed the widespread presence of Ca. A. phytopathogenicus in the western part of Switzerland (Figure 3B). All SBR-infected fields were positive for a least one VY virus, and most of them were positive for both BYV and BChV (Appendix A). This suggests that a combination of these pathogens has led to the strong yield reductions recorded at the end of the season (Figure 1). Nevertheless, how VY and SBR modulate each other in terms of symptom severity, vector fitness and prevalence deserves more scrutiny. The Stolbur phytoplasma, initially thought to be part of the SBR syndrome [8], was only detected in two samples from one site, supporting a marginal effect on yields.

Expansion of the SBR-infected area is likely led by the migration of *P. leporinus*, as specimens of this species were found in all inspected sites, and generally had a high rate of bacterial infection (Table 3). It can be thus assumed that the SBR epidemiology in Switzerland is similar to the situation described several years ago in France, where emergence of SBR has been indeed linked to outbreaks of *P. leporinus* [44,45]. This cixiid is naturally present in all Europe [46], but seems to particularly thrive in the crop rotation system consisting of sugar beet followed by winter wheat [47], which is a typical practice in Switzerland.

Results from the SBR greenhouse assay revealed the presence of the bacterium in the petiole of leaves on which the insects had fed, but also in the taproot and rootlets (Figure 7D). Along with vertical transmission, this underground tropism explains how *P. leporinus* nymphs can acquire the bacterium as previously documented [45]. The bacterium does not seem to invade emerging leaves, despite severe symptoms observed on these tissues (Figure 7B). As the taproot is easy to sample and exhibits typical vasculature symptoms, this tissue should be used for future screening. In parallel, an important root biomass reduction was evidenced in the assay (Figure 7C). However, this does not reflect natural conditions in which sugar yield is reduced but root biomass is rarely affected [48]. This difference is likely due to the fact that plants were inoculated at a young age, while this process occurs after several months of growth in fields [9]. Nevertheless, this also indicates that higher yield reductions can be feared should field inoculations occur earlier in the season.

Unlike VY viruses that can be easily maintained in the laboratory via aphids, the SBR pathosystem is currently challenging to study. So far, inoculation procedures have been limited by the capture of adults of *P. leporinus* emerging for a couple of weeks in the summer [9]. Furthermore, the latency period is quite long for sugar beet. However, future fundamental studies will be aided by the use of *C. roseus* as this model species, which develops symptoms faster than sugar beet and enables propagation of the bacterium by grafting (Figure 7E).

Since 2020, several projects have been implemented in Switzerland to limit future outbreaks of VY and SBR. Local field assays are being performed to identify tolerant and resistant cultivars, and similar work is being carried on larger germplasms at seed companies. In parallel to plant traits, changes in agriculture practices are being evaluated. For instance, replacement of wheat with barley in crop rotation as well as reduced tilling were shown to reduce field population of *P. leporinus* in France [49]. Flowering strips and intercropping in fields are being tested as well because they could increase the populations of natural predators of vectors. Furthermore, control is also being assessed with mineral oils, kaolinite and biocontrol agents. In particular, tests are being carried with several strains of *Pseudomonas* spp. given that these were shown to efficiently control aphids on sugar beet [50]. In parallel, entomopathogenic fungi and nematods have been used against *Hyalesthes obsoletus*, the cixiid vector of Bois noir [51,52], motivating the search for similar organisms to control *P. leporinus*.

## 4. Materials and Methods

### 4.1. Plant Sampling and Nucleic acid Extraction

In October 2020, sugar beet fields from various locations in Switzerland (Appendix A) were chosen for a screening targeting VY viruses and Ca. A. phytopathogenicus based on the presence of typical yellowing symptoms. A total of 98 leaf samples from individual symptomatic plants were obtained from 34 growing sites and stored at 4 °C until further analysis. For a subset of 26 sites (56 samples), taproot tips were collected in addition to leaf samples and stored in similar conditions.

For nucleic acids extraction, 0.5 g of plant material was first ground in 6 mL of extraction buffer (3% CTAB, 1.4 M NaCl, 25 mM EDTA, 1 M Tris-HCl) using the Homex 6 homogenizer (Bioreba, CH-4153 Reinach, Switzerland). Two ml of the homogenate were collected and clarified by centrifugation at 3200 rpm for 10 min. Next, 900 μL of the supernatant was transferred into a new tube and incubated for 30 min at 65 °C in the presence of 2 μL β-mercaptoethanol. One vol of chloroform: isoamyl alcool (24:1) was added and the resulting mix was vortexed for 5 sec. Following centrifugation at 5000 rpm for 5 min, 700 μL of the supernatant was transferred into a new tube. One vol of isopropanol was added and the mix was incubated for 30 min at −20 °C. After centrifugation at 14,000 rpm for 2 min, the supernatant was discarded, the pellet was washed with EtOH 70% and eventually resuspended in 100 μL nuclease-free water.

### 4.2. (RT-PCR Detection

For VY viruses, samples extracted from leaves were analyzed by one-step reverse transcription and polymerase chain reaction (RT-PCR). Reaction mixes combined AMV reverse transcriptase and GoTaq DNA polymerase (Promega, 53711-Madison, WI, USA). Specific primer pairs targeting the five VY viruses were used (Table 1). For Ca. A. phytopathogenicus, samples from roots were tested by PCR using the GoTaq DNA polymerase in combination with primer pair SpotF/R_SBR (Table 1). 

Reaction mixes were performed as described in Appendix A. Conditions for the RT were 48 °C for 45 min. Conditions for the PCR were as follows: 94 °C for 2 min followed by 35 cycles of 94 °C for 30 sec, 55 °C for 30 sec and 72 °C for 1 min 30 sec, and a final step at 72 °C for 10 min. Amplicons were analyzed by agarose gel electrophoresis.

### 4.3. Maps

Coordinates (LV95) of sampling sites were recovered using Koordinator (https://tools.retorte.ch/map/ accessed on 1 February 2022). Diagnostic data were projected on a map of Switzerland using the R package “ggswissmaps”.

### 4.4. Sequencing of Viral Genomes

For 48 and 46 samples positive to BYV or BChV, respectively, a small genomic fragment was amplified using the primers used for RT-PCR diagnostic (Table 1) and sequenced by Sanger sequencing at Fasteris (CH-1228 Plan-les-Ouates, Switzerland).

Eight leaf samples that tested positive for BChV and/or BYV (Appendix A) by RT-PCR were selected for HTS of VANA. Briefly, leaves were first ground in liquid N_2_ and mixed with cold extraction buffer (0.5 M Tris, pH 8.2, 0.2% β-mercaptoethanol, 0.5% bentonite, 4% polyclar). The mix was agitated for 20 min on ice and then filtered through a layer of cheesecloth. Next, the mix was clarified for 20 min at 4000 rpm. The supernatant was collected and deposited on a 20% sucrose cushion. After 2 h of ultracentrifugation at 40,000 rpm (5 °C), the obtained pellet was suspended in HBSS buffer [53]. Resuspended virions were then treated with bovine pancreas DNase I and RNase A (Euromedex). RNA was extracted from virions using the RNeasy kit (Qiagen) and quantify by Qubit fluorometer. Next, a cDNA library was prepared according to Maclot et al. [53] and sent to Fasteris for HTS. Illumina sequencing was performed using a NexSeq 2 × 150 Mid-Ouput flow cell. Following demultiplexing, the obtained FastQ reads were filtered via Trimmomatic [54] and used for de novo assembly using Spades [55]. The obtained contigs were used as reference for re-mapping reads using Bowtie2 [56]. When gaps were still present, viral genomes were completed using PCR followed by Sanger sequencing. The sequences of three near full-length BMYV genomes were obtained by Sanger sequencing using primers based on conserved regions found in previously published genomes.

### 4.5. Bioinformatics Analyses

Viral genomes were visualized on UGene [57] and Geneious prime version 2022.1.1. Related viral genomes were downloaded from NCBI and aligned using MUSCLE [58]. Phylogenetic analyses were performed as previously described [59]. ML trees were rooted using close viral sequences as outgroups: *Carnation yellow fleck virus* (Genbank acc. number GU234167.1) for BYV, BMYV-Moosseedorf for BChV and BChV-Villiger for BMYV. Pairwise matrices were obtained with the SDT software version 1.2 [60]. Genetic diversity was analyzed in R version 4.1.0 using the packages “Ape” and “vcfR”.

Haplotype networks were constructed using the packages “Ape” and “Pegas”. In addition to the 48 BYV and 46 BChV newly sequenced fragments and full genomes from Swiss isolates, additional sequences were gathered from GenBank (Appendix A).

### 4.6. Aphids Monitoring

Since 1983, winged aphids have been monitored at the Agroscope suction trap (Changins, Switzerland) every day from April to July. Specimens of *M. persicae* and *A. fabae* are scored individually based on morphological identification. Data for 2020 were compared with data collected from 2000–2019. Plots were generated on R.

### 4.7. SBR Vector Monitoring

Flying insects were collected using a SH 86 suction device (Stihl) in different SBR-infected fields in the summer 2020 (Table 3). Insects were sorted based on morphology [61], and placed in EtOH at −20 °C until further analysis. After CTAB extraction, DNA samples from *P. leporinus* were tested for the presence of Ca. A. phytopathogenicus by PCR using the primer pair Alb1/Oliv1 (Table 1).

### 4.8. Aphid-Mediated Inoculation

Specimens of *M. persicae* were collected from a rapeseed field near Changins and were propagated on Chinese cabbage (*Brassica rapa* subsp. *pekinensis*) in a growth chamber (15 °C, 75% relative humidity, 16 h/8 h light/dark regime). Plants of sugar beet that tested positive for either BYV or BChV were recovered from Swiss fields and were used as virus source. Virus acquisition was performed by placing aphids on a piece of symptomatic leaf in a petri dish for 8 h for BYV or 48 h for BChV. 

After acquisition, five aphids were placed on one leaf per plant in a mesh-covered plastic cage for 48 h. Aphids were then collected and leaves were carefully washed. Non-infected plants kept in the same conditions were used as controls. To further minimize the risk of aphid spread after infection, plants were treated with the insecticide Tepekki (0.1 g/L) according to the manufacturer’s instructions.

### 4.9. Cixiid-Mediated Inoculation and Grafting

Twenty adults of *P. leporinus* were collected in a heavily infected sugar beet field at La Rippe in 2021. Insects were placed in an insect-proof cage containing ten 3-week old sugar beet plants (cv. Strauss). In parallel, ten plants were grown in the same conditions without insect and were used as controls. After 2 weeks, insects were collected and plants were removed from the cage and repotted in bigger containers allowing full development. Insects were analyzed for the presence of Ca. A. phytopathogenicus by PCR as described before. A similar methodology was followed for the inoculation of four Madagascar periwinkles. For this species, one plant symptomatic and PCR-positive was then used for a graft transmission. Briefly, stem parts were grafted onto the stems of four young periwinkles, and plastics bags were placed on plants to maintain high humidity. After one week, bags were removed and plants were then daily checked for symptoms. After three weeks, plants were tested by PCR for the presence of the bacteria as described above. 

### 4.10. qPCR Analyses

At the end of the bioassays for VY viruses and SBR, nucleic acids were extracted from different tissues using the CTAB method as mentioned above. For VY viruses, quantification was performed using the GoTaq probe 1-step RT-qPCR system (Promega). For Ca. A. phytopathogenicus, quantification was performed using the GoTaq probe qPCR master mix (Promega). Data were analyzed on the CFX96 Real time system (Biorad). Mix composition is described in Appendix A.

For BYV and BChV, the primer pairs BYV-HEX F/R and BChV-FAM F/R were used in combination with respective probes (Table 3). The primers were designed against the viral genes ORF1a for BYV and ORF0/ORF1-2 for BChV. As internal control, the primers COX-ROX F/R targeting sugar beet *cytochrome oxidase* (acc. number NC_002511.2) were designed in combination with the probe COX-PROX (Table 2). Reaction efficiencies, sensitivity and limits of detection were measured using RNA samples diluted in healthy beet extract for both duplex (i.e., one virus and the plant control) and triplex (i.e., two viruses and the plant control) reactions. Due to presumably high amounts of inhibitors present after nucleic acids extraction, all samples were diluted 10-fold in water prior analysis. Thermocycler conditions were as follows: 50 °C for 20 min then 95°C for 5 min followed by 42 cycles of 95 °C for 15 s and 60 °C for 30 s.

For Ca. A. phytopathogenicus, the primer pair SpoTSBR_F/R and the associated probe (Table 2) were designed based on recent work on Ca. P. fragariae [62]. Reaction efficiencies, sensitivity and limits of detection were measured using a recombinant pGemT-Easy plasmid harboring the target sequence, in serial dilution in healthy beet extract. As internal control, the primers GluA3F/R targeting sugar beet *glutamine synthase* gene were used in combination with the probe GluD1 (Table 2). Thermocycler conditions were as follows: 95 °C for 2 min followed by 40 cycles of 94 °C for 15 sec and 60 °C for 30 sec.

## 5. Conclusions

This work highlighted a combination of re-emerging VY and high SBR prevalence in 2020 in western cantons of Switzerland, where an unprecedented sugar yield reduction was also observed. Along with increasing frequencies of VY outbreaks expected with climate change [63], the continuous expansion of the SBR-contaminated area places the whole Swiss sugar industry in a difficult position. Nevertheless, several eco-friendly control measures for both diseases are being evaluated. We now have robust quantification tools that will be of great help when assessing the effectiveness of these measures, and will also be highly useful for future monitoring and cultivars trials.

## Figures and Tables

**Figure 1 pathogens-11-00885-f001:**
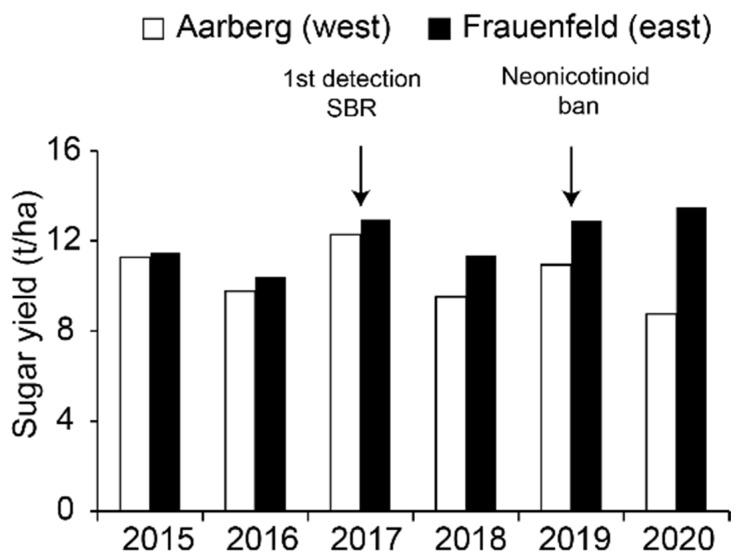
Evolution of sugar yields from sugar beet at Aarberg and Frauenfeld from 2015 to 2020. Data were reproduced from the Fédération Suisse des Betteraviers (FSB), with permission. Bars are colored according to the factory locations.

**Figure 2 pathogens-11-00885-f002:**
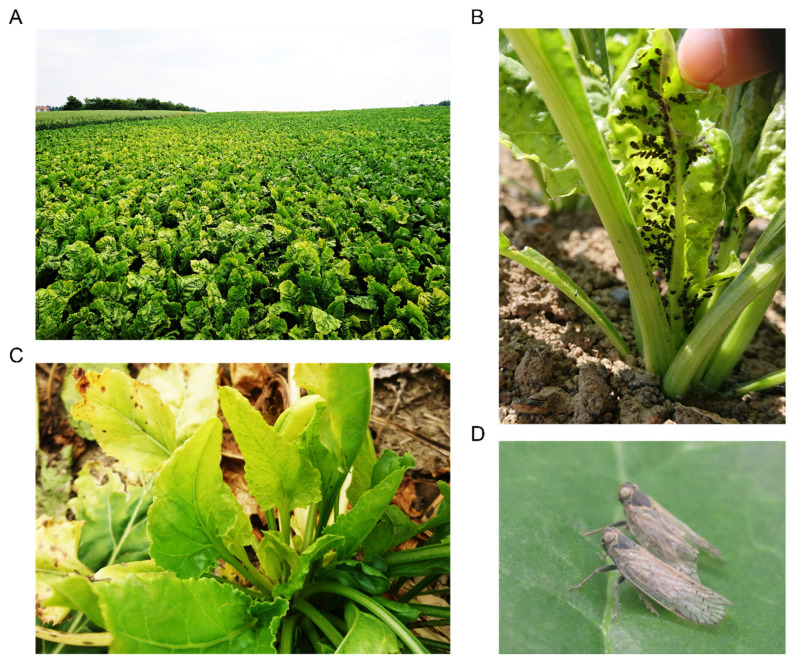
Symptoms and vectors of VY and SBR in Switzerland. (**A**) VY-associated yellowing patches in a field at Aeschi (canton of Solothurn); (**B**) heavy colonization of a young sugar beet by specimens of *A. fabae* in Penthalaz (canton of Vaud); (**C**) SBR-associated deformation of emerging leaves at Oulens (canton of Vaud); (**D**) two adults of *P. leporinus* in Changins (canton of Vaud).

**Figure 3 pathogens-11-00885-f003:**
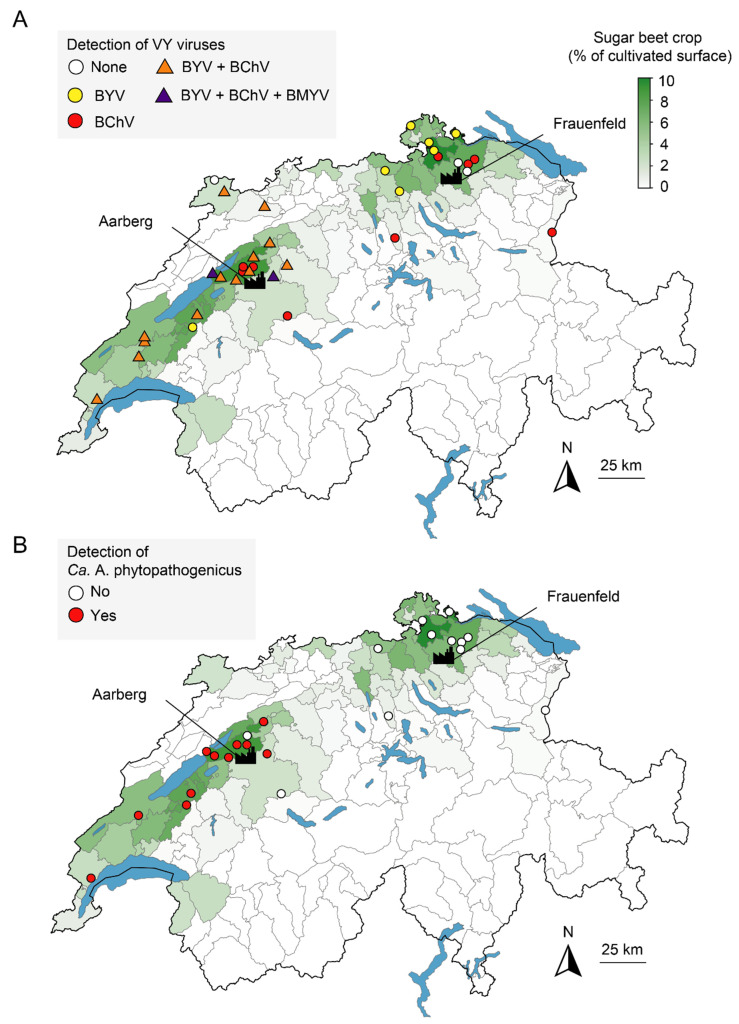
Detection of the causal agents of VY (**A**) and SBR (**B**) in Switzerland in 2020. Points indicate the location of sampling and are colored according to the detected pathogen. For VY, triangles indicate the presence of multiple viruses. Data for relative density of sugar beet crops (in green) were reproduced from the Office Fédéral de la Statistique with permission. The location of the factories of Aarberg and Frauenfeld is indicated.

**Figure 4 pathogens-11-00885-f004:**
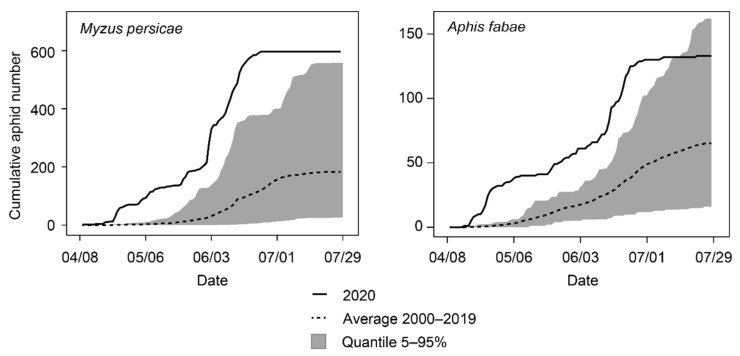
Cumulative numbers for alate specimens of two VY aphid vectors: *M. persicae* (**left**) and *A. faba* (**right**) caught at the Agroscope suction trap in 2020 in comparison with the average 2000–2019.

**Figure 5 pathogens-11-00885-f005:**
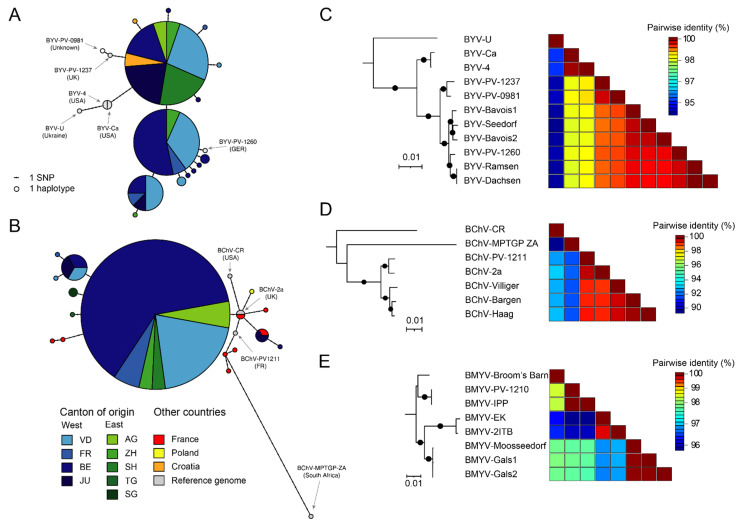
Genetic and genomic diversity of Swiss VY viruses. (**A**,**B**) Haplotype networks of BYV (A, *n* = 48) and BChV (B, *n* = 46). Additional homologous sequences from other countries were obtained from GenBank. Circle size indicate the number of sequences belonging to a given haplotype, and colors refer to the origin. AG, Aargau; BE, Bern; FR, Fribourg; JU, Jura; SG, St-Gallen; SH, Schaffausen; TG, Thurgau; VD, Vaud; ZH, Zürich. Reference sequences from full-length genomes are colored in grey. The number of variants separating two haplotypes are shown as ticks on the connecting lines. (**C**–**E**) Maximum-likelihood phylogenetic trees and associated pairwise identity matrix for genomes of BYV (**C**), BChV (**D**) and BMYV (**E**). Substitution models were as follows: TPM2u+F+I for BVY, TM2+F+I for BChV and TN+F+I for BMYV. A black circle on the branch indicates bootstrap support > 80%. Tree scales are given in substitution per site.

**Figure 6 pathogens-11-00885-f006:**
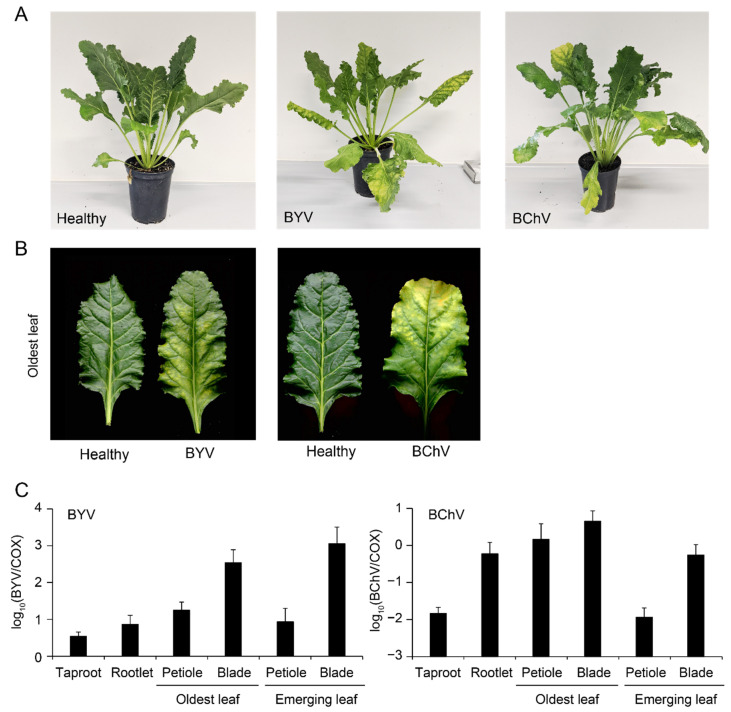
BYV and BChV infection in greenhouse conditions. (**A**) Sugar beet non-inoculated (**left**) or exhibiting symptoms of BYV (**center**) or BChV (**right**) at 60 dpi; (**B**) yellowing on the oldest leaves of plants shown in **A**; (**C**) quantification of BYV (**left**) or BChV (**right**) by duplex RT-qPCR in different tissues relative to plant cytochrome oxidase (COX). Error bars correspond to standard deviations (*n* = 6).

**Figure 7 pathogens-11-00885-f007:**
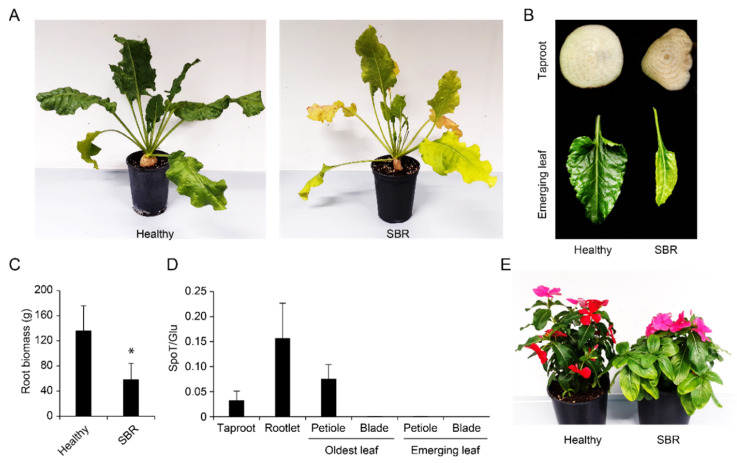
SBR infection in greenhouse conditions. (**A**) Sugar beet non-inoculated (**left**) or exhibiting SBR symptoms (**right**) at 90 dpi; (**B**) taproot cross section (**top**) and emerging leaf (**bottom**) from a healthy control (**left**) or a SBR-infected plant (right); (**C**) fresh root biomass for healthy controls (*n* = 10) or SBR-infected plants (*n* = 4); (**D**) quantification of Ca. A. phytopathogenicus DNA (SpoT) relative to plant DNA (Glu) in different tissues (*n* = 4); (**E**) yellowing on a grafted SBR-infected periwinkle (**right**) in comparison with a healthy control (**left**). Error bars correspond to standard deviations. “*” indicates *p* < 0.05 (Welch’s t-test).

**Table 1 pathogens-11-00885-t001:** Primers used for (RT-)PCR analyses.

Names	Sequences	Targets	Ref.
BYV-2a	CTATTCGGACCACCCTGCG	BYV	[14]
BYV-2b	AGGAGAATACGCTGAACGAGTGAT		
MPxBM+	ATGCAATTTCAGCTTAAAACAAACA	BMYV	[15]
MPxBM-	GCTTTCAGCACACCATACTG		
MpxBC+	ATGAACTTTGAGATTTGCTTC	BChV	
MpxBC-	TGACAGCTTTTTCTGAAGAG		
BWYV-f4972	TCTGAAGGTCGGTGAAGGAGG	BWYV	[16]
BWYV-r56	ACAATGGGTACCACTCGTTGC		
BtMV-LN 26	GACACTCAGAACTATCTCGACGAAG	BtMV	[17]
BtMV-LN 27	CACTCTGTAATGTGGAACAACTC		
SpotF_SBR	CCTATTAGAACTTCAACAAAGTGCAGG	*Ca.* A. phytopathogenicus	[10]
SporR_SBR	CGAAGGTAGAATTATCGAATTGCCT		
Alb1	GCCAGCGAGTAGAGTCGG		[18]
Oliv1	GCTGACTGCGCACCTACGC		
Stamp-F	GTAGGTTTTGGATGTTTTAAG	*Ca.* P. solani	[19]
Stamp-R0	AAATAAAAGAACAAGTATA-GACGA		
Stamp-F1	TTCTTTAAACACACCAAGAC		
Stamp-R1	AAGCCAGAATTTAATCTAGC		
rbcL_1	TTGGCAGCATTYCGAGTAACTCC	Plant rbcL	[20]
rbcL_b	AACCYTCTTCAAAAAGGTC		

**Table 2 pathogens-11-00885-t002:** Primers and probes used for the (RT-)qPCR analyses.

Names	Sequences	Targets	Ref.
SpoTSBR_F	AGGTCATGCTTGTGTTGGAG	*Ca.* A. phytopathogenicus	This study
SpoTSBR_R	TGGTCTGCCCGGTATTTAAC		
SpoTSBR-FAM-LNA	FAM-TGAGACA*AAG*GATA*AGGTTGGCGA-BHQ-1		
Glua3F	GACCTCCATATTACTGAAAGGAAG	Sugar beet	[36]
Glua3R	GAGTAATTGCTCCATCCTGTTCA		
GluD1	HEX-CTACGAAGTTTAAAGTATGTGCCGCTC-BHQ-2		
BYV-2673 F	ATAATGTTGACTCCCGCAT	BYV	This study
BYV-2773 R	GCTGAACGAGTGATTTTGTAG		
BYV-2726 P-HEX	HEX-TACGAAAACCGAATGGGCG-BHQ1		
BChV-164 F	TGGACACCAAGTTGAACAGT	BChV	
BChV-273 R	GTGTAAAGGGCCTGTCACCA		
BChV-221 P-FAM	FAM-TTGGAGCTCTTGGGACCATG-BHQ1		
COX-F	CGTCGCATTCCCGATTATCCA	Sugar beet	
COX-R	CAACTACAGAGATATAAGAGC		
COX-P-ROX	ROX-TGCTTACGCAGGATGGAATGCCCT-BHQ2		

‘*’ corresponds to LNA modified bases.

**Table 3 pathogens-11-00885-t003:** Field-collected adults of *P. leporinus* tested for the presence of Ca. A. phytopathogenicus.

Date	Location	Number of Insects PCR-Positive/Collected
06/16	Yens	7/7
06/20	Changins	3/3
06/20	Gilly	6/12
06/20	Cornaux	1/2
06/26	Alchenflüh	0/1
06/29	Penthalaz	8/10
07/01	Selzach	1/3
07/08	Zollikofen	3/7

## Data Availability

Not applicable.

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
