# Peer review of "Virus Yellows and Syndrome “Basses Richesses” in Western Switzerland: A Dramatic 2020 Season Calls for Urgent Control Measures"

_pathogens, 2022, doi:10.3390/pathogens11080885_

Round 1

Reviewer 1 Report

The virus yellows (VY) and syndrome “basses richesses” (SBR) caused major loss of sugar beet yields. VY is caused by aphid-transmitted RNA viruses, SBR is caused by the cixiid-transmitted γ-proteobacterium Candidatus Arsenophonus phytopathogenicus. In this paper, the author analyzed the disease samples and screened the causal agent for VY and SBR. The results showed that BYV and BChV are together with SBR in the west, appeared frequently nationwide. Further the author developed a quantification tools for BYV, BChV and Ca. A. However, the author did many surveys and investigation on the plant viruses (including their genetic diversity, detection, and quantification), and little work was done on the causal agent of SBR. The mechanisms underlying the linkage of the VY and SBR in the same field should be further explored. I suggest to add the contents about the mutual synergism of BYV, BChV, and Ca. A, and further to discuss the effects to the symptoms appeared on sugar beet in the field and establish the linkage between these pathogens and symptoms.

Major revisions:

1.       The VY caused by the five positive-sense RNA viruses, in present study, the author just analyzed the two viruses, and others have not systemic analyzed. It should add these.

2.       According the investigation results, whether the causal agent of VY could be narrow to the detected two viruses, BYV and BChV.

3.       In Figure 6, it should add a picture the show the mutual synergism of BYV and BChV in Fig 6B.

4.       In Figure 6C, it should add the corresponding quantification mutual synergism of BYV and BChV.

Minor revisions

1.       In Line 16, “…γ-proteobacterium Candidatus Arsenophonus phytopathogenicus…”, the “Candidatus Arsenophonus” should be italic.

2.       The Figure 1, in line 69 to 79, the format was wrong, and lead to the figure disorganized. The figure legend not show what does the blank and the black column represent. It should be explained in the figure legend.

3.       Figure 2, all the pictures should add the frame, and should add a picture show that are healthy for the control to the Fig. 2A in a large view. It also should have a big size picture to exhibit the Aphis fabae.

4.       Figure 6, It should add a picture the show the mutual synergism of BYV and BChV in Fig 6B.

5.       The reference should be reorganized according to the format of the pathogens.

6.       Table 3, the target should be revised to the PCR amplification fragment in detail. And the host genes (Glua3 and COX) sequences should be supplied in the attached files.

Reviewer 2 Report

I think it is an interesting paper, there is good coherence between the experiments and the paper is written clearly. However, I have some minor questions/remarks:

1. Is information available on the inclusivity of the used primers for virus detection, are different genotypes/variants of the viruses detected? Moreover, are BWYV and BtMV not present in the surveyed regions or where they just not detected?

2. A bit related to point 1. Was only a reference based sequence analysis used in the HTS/VANA data or was the presence of other viruses also studied? Are other viruses present?

 3. L85-86 and paragraph 5.4. Very nice that you managed to obtain 3 (near?) complete genome sequences of BMYV! Did you find any subgenomic sequences like reported for other poleroviruses?

4. L191 “, advocating for complete sequence”. Why do you assume that the online available sequences are complete? For eg MW274719 which is 15468 base pairs long is indicated that it is a partial sequence. Therefore I would like to see this statement removed or nuanced (eg to near complete).

5. L266-L283. L273, were the plants without symptoms tested too? Does an infection always cause symptoms? Maybe it is obvious, but was the qPCR described in paragraph L279-283 applied to the infected plants only? L282-283, are the healthy plants the control plants or the 6 without symptoms?

6. For how many plants was the quantification done in fig 7D? I assume also n=4?

7. The conclusion. The first sentence stated that there was a severe impact. I cannot find any data in the paper supporting this statement. Was is yield loss or only affected leaves, how severe was it? Furthermore, L385-387, is there a reference or data to support that the eco-friendly control measures show promising results? I agree that the developed tests can be very useful for future monitoring.

8. Paragraph 5.1. Could you provide some more details on the sampling? If I read it correctly, sampling was done of symptomatic plants only. Were root samples taken from the same plants as the leaf blades (there are 98 leaf samples, 56 root samples)? Furthermore, are individual plants sampled or is tissue from multiple plants pooled?

9. L437, FastaQ of FastQ reads?

10. Paragraph 5.2, L408-417. Please add the PCR mix composition and cycling conditions or refer to the reference if the conditions are the same.

11. Paragraph 5.9, L 488-504. Please add the PCR mix composition (eg primer concentration) or refer to a reference.

12. Figure S1. It might be nice to add the number of sites to the caption/figure

13. Figure S4. To be complete maybe write Madagaskar periwinkle. Was an internal control/housekeeping gene included in the RT-PCRs? Were the asymptomatic plants not infected or did the extraction not work?

Round 2

Reviewer 1 Report

I have no other suggestions, thanks!